# EMBODIED GRAPH: A RELATIONAL WORLD MODEL

## ABSTRACT

World models empower embodied exploration by building internal representations and predicting future states of the environment. Nevertheless, existing approaches merely rely on *dense* supervisions such as pixel-level signals, failing to distinguish task-relevant information from task-irrelevant information during exploration, where task-relevant information is normally *sparse*. As such, this dense-supervision design suffers from suboptimal exploration due to the noisy task-irrelevant information. To address this issue, in this paper, we propose **Embodied Graph**, a sparse-supervision design capable of capturing the *sparse* task-relevant information for embodied exploration, to the best of our knowledge, for the first time. However, the proposed **Embodied Graph** remains unexplored and imposes three challenges: 1) How to use an embodied graph to model the environment dynamics? 2) How to update the embodied graph in a dynamic environment? 3) How to define and learn graph-grounded actions and policies during explorations? To solve these challenges, we propose to instantiate **Embodied Graph** as a **R**elational **W**orld **M**odel (**RWM**) for embodied tasks execution. Specifically, we first design and formalize the embodied graph, incorporating definitions of nodes and edges and extending the concept of interactions. Based on this formulation, the instantiated **RWM** is able to serve as a hierarchical architecture consisting of high-level RWM and low-level RWM. On the one hand, the high-level RWM includes: (i) an embodied dynamic graph constructor that continually updates nodes and edges based on reachability and frontier discovery; (ii) a graph-guided macro-action generator that nominates exploratory macro-action candidates by jointly balancing exploration gain, operational cost, and potential risks. On the other hand, the low-level RWM integrates a plug-and-play behavioral model that executes the selected macro-actions. Extensive experiments over Minecraft and Atari demonstrate the effectiveness of our proposed **RWM** model in significantly outperforming the state-of-the-art baseline methods. In particular for Minecraft, among all the comparative approaches, our proposed **RWM** is the only one capable of achieving the final goal within the given budge.

## 1 INTRODUCTION

Exploration is fundamental for embodied agents to solve tasks in dynamic open environments, such as manipulation Ferraro et al. (2025); Lin et al. (2025); Noseworthy et al. (2025), navigation Zhi et al. (2025); Wen et al. (2025); Bar et al. (2025), novel material discovery Reddy & Shojaee (2025); Shahzad et al. (2024); Wan et al. (2025), etc. By exploration, embodied agents can actively discover critical environment states and update their strategies, which are essential for accomplishing tasks with sparse rewards. World model has emerged as a promising approach to achieve effective exploration by building internal representations and predicting future states of the environment for decision-making (Li et al., 2025; Hafner et al., 2025; Ren et al., 2025).

Nevertheless, existing world models merely rely on *dense* supervisions such as pixel-level signals, failing to distinguish task-relevant information from task-irrelevant information during exploration, where task-relevant information is normally *sparse*. This oversight leads to *imagination drift*, where prediction errors accumulate over long rollouts due to the noisy task-irrelevant information, progressively corrupting the imagined trajectories and shortening the effective imagination horizon. For instance, as illustrated in Fig.2, minor initial inaccuracies (e.g., smoothing a sandbar) can compound into severe deviations (e.g., hallucinating an oasis), ultimately leading the agent toward implausible

states. As such, this dense-supervision design suffers from suboptimal exploration due to the noisy task-irrelevant information.

To address this issue, we propose **Embodied Graph**, a novel sparse-supervision design capable of capturing the *sparse* task-relevant environmental dynamics information for embodied exploration. Unlike dense representations, graphs offer a structured and compact abstraction of entities and their relations, which can naturally filter out irrelevant details and focus on critical interactions. However, leveraging graph representations for embodied exploration introduces several challenges: (1) Although graphs can compactly represent entities and their relations, it will be challenging to model the environment using an embodied graph. (2) Since environments are open and dynamic, entities and relations may appear, disappear, or alter their attributes, thus it is also challenging to update the graph structure online while maintaining consistency and stability. (3) Different from the primitive-level actions used by existing methods, designing macro-actions requires rethinking how actions are represented and executed so that they remain both expressive and effective for control, therefore it is challenging to define and learn graph-grounded macro-actions as well.

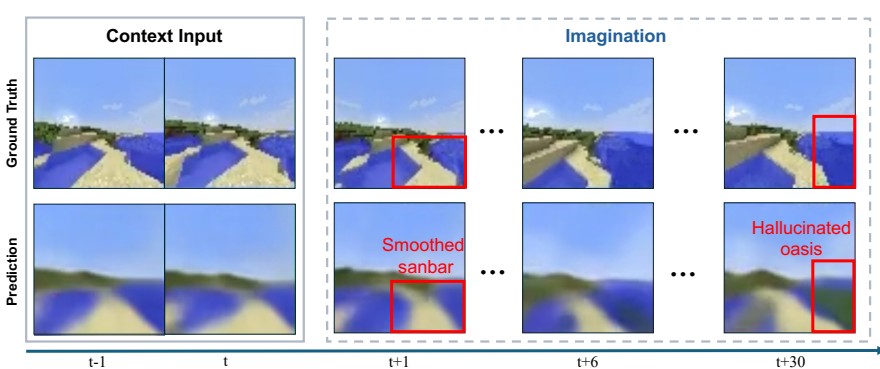

Figure 1: Task-irrelevant information in the dense observations will lead to *imagination drifts*. Minor initial inaccuracies (e.g., smoothing a sandbar) can compound into severe deviations (e.g., hallucinating an oasis), ultimately leading the agent toward implausible states.

To tackle these challenges, we propose to instantiate **Embodied Graph** as a **R**elational **W**orld **M**odel (**RWM**) for embodied tasks execution. Specifically, we first design and formalize the embodied graph, incorporating definitions of nodes and edges and extending the concept of interactions. Among them, the nodes represent agents and entities; the edges represent node relations to simulate and learn the intrinsic environmental abstractions; the interactions are the actions from the agents to their target node. Based on this formulation, the instantiated **RWM** is able to serve as a hierarchical architecture consisting of high-level RWM and low-level RWM. On the one hand, the high-level RWM includes: (i) an embodied dynamic graph constructor that continually updates nodes and edges based on reachability and frontier discovery; (ii) a graph-guided macro-action generator that nominates exploratory macro-action candidates by jointly balancing exploration gain, operational cost, and potential risks. On the other hand, the low-level RWM integrates a plug-and-play behavioral model that executes the selected macro-actions. This hierarchical architecture allows the high-level model to focus on global relational dynamics construction while the low-level model handles local embodied behavior execution, thereby improving the fidelity and horizon of imagination.

Extensive experiments in Minecraft (Hill et al., 2023) and Atari (Bellemare et al., 2013) demonstrate that our proposed **RWM** model is able to significantly outperform state-of-the-art baselines, particularly in tasks requiring long-horizon exploration and sparse reward handling, such as collecting diamonds. The results validate that our sparse-supervision design effectively mitigates imagination drift and enhances exploration efficiency. In summary, we make the following contributions:

- To the best of our knowledge, we propose Embodied Graph, the first graph-based paradigm for modeling environment dynamics in embodied exploration.
- We propose a Relational World Model, a hierarchical framework that integrates a high-level global relational dynamics construction with a plug-and-play low-level local embodied behavior execution, supported by an online embodied dynamic graph constructor and a graph-guided macro-action generator.

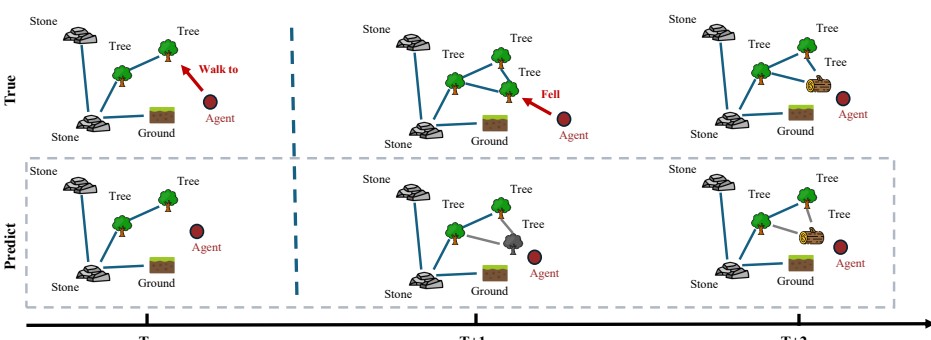

Figure 2: The prediction task of the Embodied Graph. With a graph observation, conditioned on an action of the target node, the Embodied Graph is trying to predict "how will the graph change".

- We demonstrate through extensive experiments that RWM achieves superior performance in challenging exploration tasks compared with existing methods.

## 2 EMBODIED GRAPH

Graph, as an abstract representation of the relational structure, is suitable to represent the environment and capture the underlying object relations. However, traditional graphs like 3D scene graphs are static structures, which can't interact with the agent. For this reason, we propose the Embodied Graph (EG) as an interactive graph that is able to capture high-level dynamics from real environments. Formally, we define the embodied graph and the corresponding task as follows.

**Embodied Graph**  Embodied Graph at time $t$ as $\mathcal{G}^t(\mathcal{V}^t, \mathcal{E}^t, \mathcal{I}^t)$, where $\mathcal{V}^t$, $\mathcal{E}^t$ and $\mathcal{I}^t$ are the sets of nodes, edges, and interactions at timestamp $t$, respectively. The embodied graph can be expressed as $\mathcal{G}(\mathcal{V}, \mathcal{E}, \mathcal{I}) = (\{\mathcal{G}^t(\mathcal{V}^t, \mathcal{E}^t, \mathcal{I}^t)\}_{t=1}^T)$, where $\mathcal{V} = \cup_{t=1}^T \mathcal{V}^t$, $\mathcal{E} = \cup_{t=1}^T \mathcal{E}^t$, $\mathcal{I} = \cup_{t=1}^T \mathcal{I}^t$.

**Node, edge sets**  Since the agent is a special kind of node, which can actively interact with others. The nodes contained in embodied graphs are heterogeneous and can be expressed as $\mathcal{V} = \{\mathcal{V}_a, \mathcal{V}_o\}$, where $\mathcal{V}_a$ are agent nodes and $\mathcal{V}_o$ are object nodes. Furthermore, edges can be classified as object-object edges, object-agent edges, and agent-agent edges (only exist in multi-agent systems), which are also heterogeneous and can be denoted as $\mathcal{E} = \{\mathcal{E}_{oo}, \mathcal{E}_{oa}(, \mathcal{E}_{aa})\}$. More specifically, both nodes and edges can involve the corresponding features $\mathcal{X} = \{\mathcal{X}_o, \mathcal{X}_a\}$ and $\mathcal{S} = \{\mathcal{S}_{oo}, \mathcal{S}_{oa}(, \mathcal{S}_{aa})\}$.

**Interaction set**  Different from traditional graphs, embodied graphs involve interactions between different nodes. Due to the heterogeneity of the nodes, similar to the edges, the interactions can be divided as $\mathcal{I} = \{\mathcal{I}_{oo}, \mathcal{I}_{oa}(, \mathcal{I}_{aa})\}$. The corresponding interaction feature is denoted as $\mathcal{R} = \{\mathcal{R}_{oo}, \mathcal{R}_{oa}(, \mathcal{R}_{aa})\}$.

**Prediction Tasks**  As shown in Fig, 2, the embodied graph is a way to model the environment and serves to predict future environments based on the interaction of the past environments, i.e. $\mathcal{G}^{t+1} = f_\theta(\mathcal{G}^{1:t}, i^t, n_a^t)$, where $f_\theta$ is the learnable dynamic network to predict the future embodied graph., $i^t \in \mathcal{I}^t, n_a^t \in \mathcal{V}^t$ is the chosen interaction and the target node with which to interact. An ego-EG of node $v$ at time $t$ is defined as $\mathcal{G}_v^t = (\mathcal{N}_v^t, \mathcal{E}_v^t, i^t)$, where $\mathcal{N}_v^t$ is the $L$-hop neighbors of node $v$ at time $t$, and $\mathcal{E}_v^t$ involve all edges between nodes in $\mathcal{N}_v^t$. The optimization objective of learning the embodied graph is defined as follows:

$$\min_\theta \mathcal{L}(f_\theta(\mathcal{G}_v^{1:t}, i^t, n_a^t), \mathcal{G}_v^{t+1}). \tag{1}$$

## 3 RELATIONAL WORLD MODEL

Guided by the **Embodied Graph** paradigm, we propose a Relational World Model (RWM), which is a hierarchical architecture that coordinates high-level relations with local details to guide exploration. Grid-based environment observations are first transformed into a dynamic graph for the high-level RWM, whereas the low-level RWM operates directly on raw images. The two tiers learn complementary aspects of dynamics: abstract, object-centric relations and fine-grained perceptual detail, respectively. The high-level world model produces plans $A$ via the relation-aware imagination, and the low-level behavior model executes primitive actions $a$ to achieve the given $A$. Real environment feedback is used to train both models. In what follows, we first introduce the high-level world model (Section 3.1), which comprises two main components, the *Dynamic Embodied Graph Constructor* (Section 3.1.1) and the *Graph-Guided Macro-Action Generator* (Section 3.1.2). The low-level behavior model is then explained in Section 3.2.

### 3.1 HIGH-LEVEL WORLD MODEL

Fig.3.1 gives an overall structure of our proposed high-level world model, which comprises two components: a dynamic embodied graph constructor to dynamically capture a graph for high-level environment perception, and a graph-guided macro-action generator to nominate feasible yet valuable areas to form the macro-action space. Conditioned on the graph perception and macro-action sets, following the Recurrent State-Space Model architecture Hafner & et al. (2025), we define the high-level world model training loss as:

$$\mathcal{L}(\phi) \doteq E_{q_\phi} \left[ \sum_{t=1}^{T} \left( \beta_{\text{pred}} \, \mathcal{L}_{\text{pred}}(\phi) + \beta_{\text{dyn}} \, \mathcal{L}_{\text{dyn}}(\phi) + \beta_{\text{rep}} \, \mathcal{L}_{\text{rep}}(\phi) \right) \right]. \tag{2}$$

Here $T$ is the length of inputs, $\phi$ are the world model parameters, $\mathcal{L}_{\text{pred}}$, $\mathcal{L}_{\text{dyn}}$, and $\mathcal{L}_{\text{rep}}$ are the prediction loss, dynamics loss and representation loss, respectively. $\beta_{\text{pred}}$, $\beta_{\text{dyn}}$, and $\beta_{\text{rep}}$ are the corresponding coefficients. The prediction loss:

$$\mathcal{L}_{\text{pred}}(\phi) \doteq \mathcal{L}_{\text{graph}} + \mathcal{L}_{\text{rew}} + \mathcal{L}_{\text{con}}, \tag{3}$$

contains the decoder loss of the graph, the reward and the continuity. Reward and continuity loss can be easily calculated through MSE loss and binary classification loss. However, the graph loss is hard to calculate: It is difficult to reconstruct a graph and evaluate the reconstruction effect. For this reason, we design several metrics to describe the graph and reconstruct these values instead to ensure that the world model actually understands what it observed. The dynamic loss and the representation loss can be calculated with the same data but training different aspects of the world model. Given a sequential model $h_t = f_\phi(h_{t-1}, z_{t-1}, a_{t-1})$, an encoder $z_t \sim q_\phi(z_t|h_t, x_t)$, and a dynamics predictor $\hat{z}_t \sim p_\phi(z_t|h_t)$, with a stop-gradient operator $sg(\cdot)$ on the representation $q_\phi(z_t|h_t, x_t)$, the dynamic loss is defined as follows:

$$\mathcal{L}_{\text{dyn}}(\phi) \doteq \max\left\{1, \, \text{KL}\left[ sg\left(q_\phi(z_t \mid h_t, x_t)\right) \, \big\| \, p_\phi(z_t \mid h_t) \right] \right\}, \tag{4}$$

minimizes the KL divergence between the predicted feature and the next stochastic representation to learn the dynamics of the environment. On the other hand, with the $sg(\cdot)$ on the dynamics predictor $p_\phi(z_t|h_t)$, we define the representation loss:

$$\mathcal{L}_{\text{rep}}(\phi) \doteq \max\left\{1, \, \text{KL}\left[ q_\phi(z_t \mid h_t, x_t) \, \big\| \, sg\left(p_\phi(z_t \mid h_t)\right) \right] \right\}, \tag{5}$$

train the encoder to make its representations become more predictable.

### 3.1.1 DYNAMIC EMBODIED GRAPH CONSTRUCTOR

The high-level world model requires structured data to learn relational dynamics. However, agents' observations, e.g., images, sensory data, or voxel observations, do not explicitly contain such structural information. For this reason, with the raw voxel observation, we propose a two-stage graph construction pipeline: dynamic history fusion and rule-based graph construction. Concretely, to fuse historical observations, we maintain a cubic region of interest (ROI) with side length $S$, centered on the agent. In each step, the voxel observations are fused into a grid map $M$ in the ROI index space. Instead of treating $M$ as an unstructured array, we derive semantic masks that indicate air ($b^{air}$),

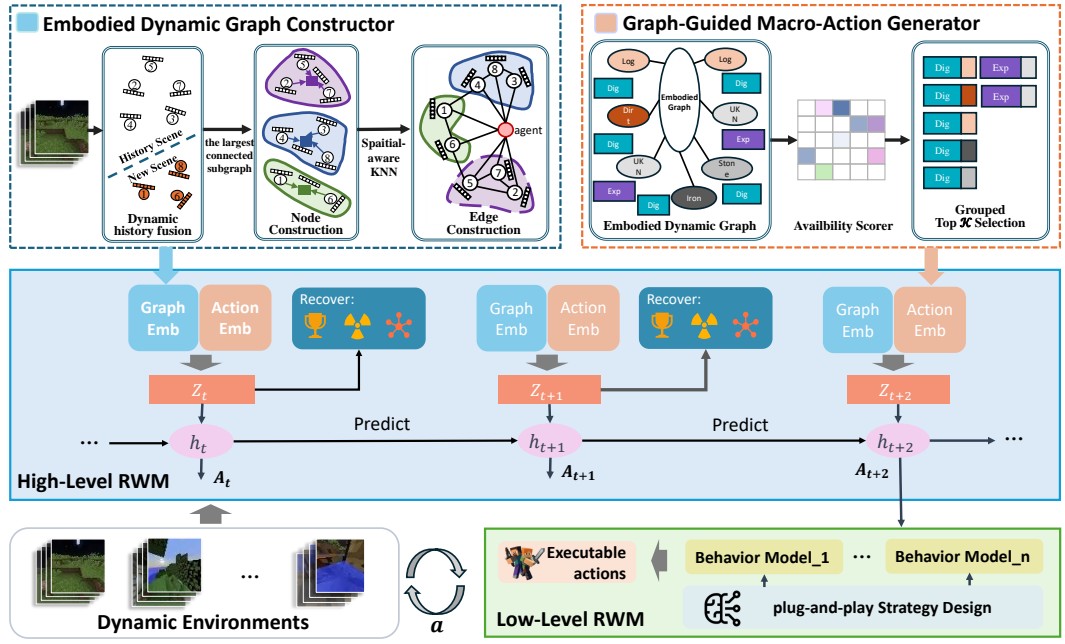

Figure 3: The Overall framework of the high-level world model, which consists of an Embodied Dynamic Graph Constructor to build and update the graph observation, and a Graph-Guided Macro-Action Generator for nominating an exploratory action set.

liquid ($b^{liq}$), diggable ($b^{dig}$), and unbreakable ($b^{unb}$) voxels. With a Y-up coordinate $x, y, z$, the *walkable region* is derived as:

$$r^{walk}_{x,y,z} = b^{air}_{x,y,z} \wedge b^{air}_{x,y+1,z} \wedge b^{solid}_{x,y-1,z} \tag{6}$$

, where $b^{solid} = b^{dig} \vee b^{unb}$, indicating locomotion affordances rather than mere occupancy. Since the agent is continuously moving, directly calculating $r^{walk}$ on the whole map is costly and unstable. Therefore, we propose to maintain an online reachable region $S_t$ through a $T$-iteration dilation on a local walkable region $r^{walk}_{loc}$. At each step $t$:

$$S_t = \bigvee_0^T (S^T_{t-1} \vee (r^{walk}_{loc} \wedge \mathcal{N}_6(S^T_{t-1})) \tag{7}$$

, where $\mathcal{N}_6(\cdot)$ is 6-neighborhood of the region. If the local walkable region has radius $\leq T$, $S_t$ equals the BFS closure.

To expose relational structure, we form frontiers and dig bands:

$$frontier = \mathcal{N}_6(b^{unk} \vee b^{dig}) \wedge S_0 \tag{8}$$

$$digband = \mathcal{N}_6(S_0) \wedge b^{dig} \tag{9}$$

, then partition the space into three agent-centric y-bands (below/around/above) and $H_b$ horizontal sectors. To reduce computational cost, each band $\times$ sector is compressed in the $y$ direction, and a 3 $\times$ 3 propagation is performed to obtain connected components $L$ on the 2-D projection, computing per-component area $A_L$, boundary contact $C_L$, centroid $(\bar{x}_L, \bar{z}_L)$, and distance $d_L$ from the agent, we form node's attributes:

$$h_{L_n} = [type_{L_n}, log(1 + A_{L_n}), C_{L_n}, (\bar{x}_L, \bar{z}_L), d_{L_n}] \tag{10}$$

Links are generated by star edges (from $S_0$ to each node) and kNN near edges among the nodes, type and distances are encoded as the edge feature.

### 3.1.2 GRAPH-GUIDED MACRO-ACTION GENERATOR

Unlike the low-level behavior model, which has a predefined primitive action space, the action on the Embodied Graph is defined as the interaction between the agent and other nodes. Therefore,

the action space can be simply defined as the node set on the embodied graph except the agent itself, or formally: $\mathbb{A}^t = \mathcal{V}^t \wedge \neg a$. However, there are two problems: i) Not all of the nodes are easy to reach, and the agent may perform badly on some very difficult actions; ii) the agent cannot learn meaningful actions on such a continuously changing action space. For these reasons, we propose a graph-guided macro-action generator to nominate an exploratory action space based on the geometrical availability. For each node $n$, contact length $c_n$ with the $S_t$, node volume $V_n$, distance from the agent $d_n$, and the node's type are used as the geometrical attributes, and the availability score can be calculated as:

$$
s_n = \begin{cases} \alpha_{fc}\,c_n + \alpha_{fv}\,v_n - \alpha_{fd}\,d_n, & t_n = \text{frontier}, \\ \alpha_{dc}\,c_n + \alpha_{dv}\,v_n - \alpha_{dd}\,d_n + \alpha_{dy}\,d_y^{\downarrow}, & t_n = \text{digband}. \end{cases} \tag{11}
$$

$\alpha$s are the corresponding coefficients to balance their importance, and we provide $d_y^{\downarrow}$ to encourage downward digging. To promote spatial dispersion, the nodes are binned into the aforementioned $H_b$ sectors. Within each type and sector, we keep the top $\mathcal{K}$ candidates by $s_n$, and padding is applied if needed. The nominated nodes and their geometrical attribute will be provided to serve as the action space. Importantly, there is no candidate information during the imagination phase, so we train an action space proposer to "imagine" the current action space for the agent to interact with.

## 3.2 LOW-LEVEL BEHAVIOR MODEL

Different from the sparsity requirement of the high-level world model, the low-level behavior model is an independent module that takes dense observations and interacts directly with the environment. According to our design, the low-level behavior model could be any of the existing policy models. Be aware that neither the observation space nor the action space of the low-level behavior model is influenced by the high-level world model, which yields strong modularity: the low-level behavior model can be developed, verified, and tuned independently, while the high-level world model can plug in and play with any of the low-level components.

## 4 EXPERIMENTS

In this section, we conduct extensive experiments to verify that our proposed Relational World Model can effectively learn the abstract relational dynamics of the environment and help the agent explore efficiently. This section is organized as follows: we introduce our experiment setup in Section 4.1, and quantitative and qualitative comparisons with baselines are given in Section 4.2.

### 4.1 EXPERIMENT SETUP

**Simulators** *Minecraft* (Hill et al., 2023) is one of the world's most-played games, drawing its players into a procedurally generated 3D sandbox. Players traverse varied biomes and cave systems, breaking blocks, managing health and hunger, and fighting hostile mobs while advancing gear. Among resources, diamond is pivotal for enabling powerful upgrades (enchanted tools/armor, obsidian mining), which makes it a central objective for most players, especially in the early-to-mid game. We also test the cross-environment generalization of our method in the Atair simulator. *Atari* (Bellemare et al., 2013) is a cutting-edge, high-fidelity simulation environment for multi-physics analysis and hardware-in-the-loop testing in aerospace and robotics.

**Task** Collecting diamonds in the open-world game Minecraft is a big challenge in artificial intelligence. Every episode in this game is a completely new 3D world, where the player needs to find and craft diamonds from scratch. We follow prior work (Hafner & et al., 2025) to increase the speed at which blocks break. To simplify the task, the agent is born with the tools to craft diamonds, so they can focus on exploring the world and finding the targets without worrying about the tools. To further validate the performance of our proposed RWM model, we performed experiments on the Atari-100k benchmark (Kaiser et al., 2020), a sample-efficiency suite built from the Atari Learning Environment (ALE). It contains 26 Atari 2600 games covering diverse dynamics (navigation, shooting, puzzle-like planning).

Table 1: Experimental results in Minecraft. Metrics: success rate (SR↑, %), success length (SL↓, steps), and survival rate (SVR↑, %). The best results are highlighted in bold and the second best are set in underline.

| Steps | Model | Stone | | | Iron | | | Gold | | | Diamond | | |
|---|---|---|---|---|---|---|---|---|---|---|---|---|---|
| | | SR | SL | SVR | SR | SL | SVR | SR | SL | SVR | SR | SL | SVR |
| 1M | PPO | 76.34 | 436 | 31.87 | 0 | – | 33.54 | 0 | – | 26.88 | 0 | – | 29.31 |
| | IMPALA | 23.24 | 1384 | 11.43 | 2.56 | 8643 | 16.22 | 0 | – | 10.02 | 0 | – | 13.22 |
| | Rainbow | 0 | – | 0.17 | 0 | – | 0.24 | 0 | – | 0.12 | 0 | – | 0.22 |
| | DreamerV3 | 97.83 | 282 | 38.67 | 24.44 | 1515 | 35.80 | 0 | – | 40.85 | 0 | – | 33.33 |
| | Ours | 91.89 | 846 | 43.28 | 36.54 | 2179 | 43.28 | 1.59 | 1592 | 48.33 | 0 | – | 44.77 |
| 2M | PPO | 81.39 | 558 | 20.74 | 0 | – | 18.37 | 0 | – | 23.39 | 0 | – | 25.47 |
| | IMPALA | 25.33 | 1497 | 12.21 | 1.7 | 8643 | 13.21 | 0 | – | 10.37 | 0 | – | 17.86 |
| | Rainbow | 1.22 | 577 | 0.37 | 0 | – | 0.73 | 0 | – | 0.12 | 0 | – | 0.21 |
| | DreamerV3 | 98.11 | 185 | 15.43 | 32.04 | 1467 | 21.43 | 2.53 | 13011 | 23.57 | 0 | – | 20.71 |
| | Ours | 94.67 | 863 | 24.29 | 39.64 | 2333 | 29.73 | 3.28 | 2694 | 30.56 | 1.49 | 4586 | 27.05 |
| 3M | PPO | 79.21 | 342 | 12.86 | 0 | – | 16.61 | 0 | – | 22.24 | 0 | – | 19.22 |
| | IMPALA | 27.05 | 1370 | 13.67 | 0.82 | 8643 | 14.42 | 0 | – | 12.75 | 0 | – | 14.33 |
| | Rainbow | 1.43 | 1248 | 0.94 | 0 | – | 0.86 | 0 | – | 0.33 | 0 | – | 0.59 |
| | DreamerV3 | 9.88 | 161 | 10.39 | 37.27 | 1652 | 16.41 | 0.14 | 13011 | 17.37 | 0 | – | 15.63 |
| | Ours | 98.22 | 544 | 17.53 | 41.57 | 2386 | 26.54 | 2.87 | 2654 | 26.90 | 4.10 | 4516 | 24.39 |
| 4M | PPO | 82.26 | 357 | 13.64 | 0 | – | 15.32 | 0 | – | 21.47 | 0 | – | 15.99 |
| | IMPALA | 34.17 | 1208 | 13.38 | 1.36 | 6960 | 13.33 | 0 | – | 14.93 | 0 | – | 15.52 |
| | Rainbow | 0.76 | 1563 | 1.24 | 0 | – | 0.77 | 0 | – | 0.54 | 0 | – | 0.88 |
| | DreamerV3 | 98.70 | 145 | 8.61 | 41.53 | 1968 | 15.29 | 1.04 | 13011 | 13.58 | 0 | – | 12.90 |
| | Ours | 98.64 | 550 | 15.23 | 45.54 | 2421 | 22.71 | 4.87 | 3516 | 22.17 | 3.05 | 4077 | 25.60 |
| 5M | PPO | 83.26 | 446 | 14.33 | 0 | – | 13.33 | 0 | – | 15.52 | 0 | – | 17.32 |
| | IMPALA | 42.15 | 1292 | 12.52 | 1.1 | 6960 | 10.21 | 0 | – | 13.31 | 0 | – | 13.33 |
| | Rainbow | 1.01 | 1487 | 1.32 | 0 | – | 1.12 | 0 | – | 0.67 | 0 | – | 0.53 |
| | DreamerV3 | 98.70 | 125 | 7.23 | 41.45 | 2218 | 12.58 | 1.79 | 13663 | 11.04 | 0 | – | 10.77 |
| | Ours | 99.23 | 498 | 14.09 | 44.69 | 2282 | 19.29 | 4.81 | 3902 | 22.53 | 3.38 | 3902 | 25.51 |

**Baselines** As our baseline, we adopt DreamerV3 (Hafner & et al., 2025), a state-of-the-art model-based reinforcement learning agent. DreamerV3 learns a latent dynamics model from high-dimensional observations and optimizes policies by imagining trajectories in the latent space, which substantially improves sample efficiency. It is also the first agent to learn to obtain diamonds in Minecraft, without human demonstrations or handcrafted curricula. This milestone highlights DreamerV3's ability to handle extremely sparse rewards and long-horizon exploration, making it a particularly strong and relevant baseline for our study. For a fair comparison, we also adopt DreamerV3 as our low-level behavior model and keep the same model size. What's more, we also include PPO(Schulman et al., 2017), IMPALA(Espeholt et al., 2018), Rainbow(Hessel et al., 2018) as our baselines to ensure fair, protocol-aligned comparison against widely used model-free and model-based approaches.

**Evaluation Metrics** To evaluate agents' performance, we choose to use the average Success Rate (SR), average Success Length (SL), and average Survival Rate (SVR) as our evaluation metrics. In particular, SR and SVR are the number of successful/surviving agents divided by the total number of agents, and SL is the earliest time that an agent finds its target. As for the Atari-100k benchmark, we report the score per game.

## 4.2 EXPERIMENT RESULTS

The experimental results are reported in Table 1. We have the following observations.

Our proposed RWM method consistently and significantly outperforms the DreamerV3 Hafner & et al. (2025) backbone. During training, both agents gradually improve their ability to collect target materials, as indicated by the increasing success rate. For relatively simple tasks, such as collecting stone and iron, their performances remain similar, since these tasks can be solved through simple random exploration without requiring a deep understanding of environment dynamics. However, the average success length reveals that RWM, by leveraging relational understanding of the environment, can locate target materials more efficiently. The performance gap becomes more pronounced on more challenging tasks, such as collecting gold and diamonds. These materials are buried deeper

Table 2: Atari-100k (400K environment steps) scores. The best results are highlighted in bold.

| Task | PPO (400K) | DreamerV3 (400K) | RWM (400K) |
|---|---|---|---|
| Alien | 276 | 1118 | **1244** |
| Amidar | 26 | 97 | **101** |
| Assault | 327 | **683** | 675 |
| Asterix | 292 | 1062 | **1079** |
| Bank Heist | 14 | **398** | 388 |
| Battle Zone | 2233 | 20300 | **22950** |
| Boxing | 3 | **82** | 74 |
| Breakout | 3 | **10** | 9 |
| Chopper Command | 1005 | 2222 | **2487** |
| Crazy Climber | 14675 | 86225 | **89132** |
| Demon Attack | 160 | 577 | **601** |
| Freeway | **2** | 0 | 0 |
| Frostbite | 127 | 3377 | **3592** |
| Gopher | 368 | 2160 | **2271** |
| Hero | 2596 | 13354 | **15310** |
| Jamesbond | 41 | 540 | **550** |
| Kangaroo | 55 | 2643 | **2896** |
| Krull | 3222 | 8171 | **8731** |
| Kung Fu Master | 2090 | 25900 | **27100** |
| Ms Pacman | 366 | 1521 | **1740** |
| Pong | -20 | **-4** | -6 |
| Private Eye | 100 | **3238** | 3120 |
| Qbert | 317 | 2921 | **3020** |
| Road Runner | 602 | 19230 | **21826** |
| Seaquest | 305 | 962 | **1103** |
| Up N Down | 1502 | 46910 | **50021** |

and associated with sparse rewards. DreamerV3 often fails to acquire useful knowledge to reach them, even if it happens to encounter them occasionally. In contrast, our proposed RWM captures abstract object relations and utilizes sparse rewards more effectively. As a result, it continues to improve its ability to discover difficult targets, a trend further corroborated by the shorter average success lengths.

Exploration in open-world environments is inherently risky, as agents may encounter hazardous situations, e.g., lava, hostile mobs, etc. Thus, beyond the objective of obtaining diamonds, it is equally critical for an agent to maintain survival throughout its exploration. To assess this, we further evaluate the agent's survival rate (SVR) during exploration, providing a complementary perspective on its overall effectiveness. Experiment results are given in Fig.4. Red and blue lines represent the survival rate of DreamerV3 and RWM, respectively. During the training process, the survival ratio continuously increases, indicating that both agents gradually learn strategies to avoid hazardous situations and maintain longer interactions. Notably, RWM achieves a much higher survival rate throughout training, suggesting that by incorporating structured world

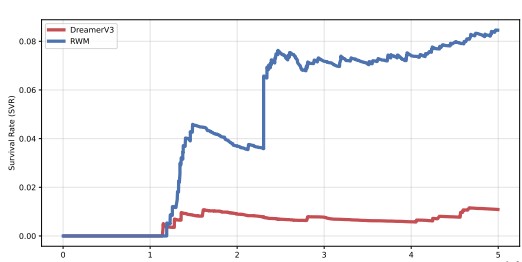

Figure 4: Comparison of the survival rate (SVR) between DreamerV3 and RWM. RWM increases its survival rate markedly after ~1.2M steps and continues to improve, whereas DreamerV3 shows only minor gains and remains below 0.015 across 0–5M steps. This suggests that structured world modeling helps the agent avoid risks and remain alive in dynamic environments.

modeling, the agent is able to distinguish the hazard environment and avoid it, which enhances robustness to environmental risks.

To further assess the generality of our approach beyond Minecraft, we evaluate RWM on the Atari-100k benchmark. Across 26 games, RWM achieves consistently higher human-normalized scores than strong model-free baselines (e.g., PPO) and the model-based DreamerV3, with the largest gains

on long-horizon, multi-goal tasks such as Crazy Climber, Road Runner, Up N Down, Battle Zone, and Hero. These results indicate that structured world modeling on object relations enables sample-efficient exploration and risk-aware decision making. We also find that, on games requiring fast reactions, like Freeway, our method is not able to perform a good result because of the simple dynamics.

## 5 RELATED WORK

**World Models.** World models empower embodied AI by building internal representations (Bruce et al., 2024; Chen et al., 2022; Robine et al., 2023; Wang et al., 2024) and making future predictions (Hafner et al., 2020; Hafner & et al., 2021; Okada & Taniguchi, 2022; Wu et al., 2022) of the external world. pixel-level and latent-level dynamics models provide a powerful tool for reasoning through imagination. The early world models (Ha & Schmidhuber, 2018) use a generative RNN to simulate future frames in a latent space. More recent work has extended world models with greater capacity and generalization. PWM (Georgiev et al., 2024) scales policy learning with large pre-trained world models across a diverse range of continuous control tasks, achieving strong performance without requiring online planning. STORM (Zhang et al., 2023) integrates stochastic latent dynamics with a transformer architecture, enhancing multi-step prediction and achieving human-level scores on Atari benchmarks. Dreamer and its successors (Hafner et al., 2020; Hafner & et al., 2021; 2025) enable agents to plan and learn from imagined trajectories, significantly improving sample efficiency and enabling long-term credit assignment. DreamerV3 notably solved the challenging "obtain diamond" task in Minecraft using pure model-based learning. However, current world model methods focus on local details while do not have high-level abstract world understanding, the dense pixel-level or latent-level observation limit their imagination length, which directly influence their exploration efficiently.

**Graphs in Embodied AI.** Due to their sparsity and superior abstraction ability, graphs have been increasingly adopted in embodied AI, providing structured inductive biases for reasoning and decision-making. Graph neural networks enable agents to encode relational information among entities and objects in the environment (Scarselli et al., 2008; Kipf & Welling, 2017; Wu et al., 2021). In embodied tasks, graph-based representations have been used to capture object-centric relations and spatial dependencies (Shen et al., 2021; Huang et al., 2022), allowing agents to generalize beyond pixel-level perception. Neural-SLAM methods (Zhang et al., 2017; Chaplot et al., 2020) combine graph structures with mapping and navigation, where nodes represent places or objects, and edges capture connectivity. Scene graph approaches (Huang et al., 2022; Wu et al., 2023) provide higher-level semantic abstraction, improving embodied agents' navigation and planning capabilities. Knowledge graphs further enhance embodied agents by injecting commonsense and affordance priors, enabling reasoning about object functionality and improving generalization to unseen scenarios (Yang et al., 2018; Wang et al., 2021; Li et al., 2024). More recent work combines transformers with graph encodings (Wu et al., 2023; Li et al., 2023), scaling relational reasoning and enabling multi-step planning in complex 3D environments. Despite their advantages, existing approaches are typically grounded in scene graphs or knowledge graphs, which only statically capture semantic or relational structures, without modeling how such attributes and relations evolve through agent–environment interaction. This limitation restricts embodied agents to passive reasoning, reducing their effectiveness in dynamic open environments.

## 6 CONCLUSION

In this paper, we propose Embodied Graph, which is a paradigm of using graph in modeling abstracted environment dynamics. Based on this paradigm, we further propose the Relational World Model (RWM), a hierarchical architecture to combine high-level relational environment dynamics and complete it with low-level details to increase agents' exploration efficiency. We first propose an embodied dynamic graph constructor to build and update a graph online to preserve rich context information of the environment. Then we develop a graph-guided macro-action generator to use geometrical information to form an exploratory action space. Finally, conditioned on the constructed graph and the candidate action space, our proposed model is able to learn abstract environment dynamics to help agents explore efficiently. Extensive experiments in Minecraft and Atari validate the great performance of RWM and the effectiveness of our design.

## ETHICS STATEMENT

This study relies solely on the publicly released Minecraft environment and assets; no human subjects, user-generated data, or personal identifiers are involved. We audited the game content and our experimental setup for ethical risks (e.g., biased or offensive material) and found none relevant to our tasks. Accordingly, we assess the risk of societal harm or unintended bias to be minimal. The work follows standard research-ethics practices and is designed to align with these principles.

## REPRODUCIBILITY STATEMENT

To help with the reproducibility, we will release our code at the publication time.

## THE USE OF LARGE LANGUAGE MODELS (LLMS)

We used LLMs as general-purpose assist tools. Specifically:

- **Code debugging and tooling:** LLMs were consulted to diagnose implementation issues (e.g., log parsing, plotting, and minor refactoring) and propose alternative snippets. All changes were reviewed, tested, and integrated by the authors.
- **Writing support:** LLMs suggested phrasing improvements and helped polish drafts (e.g., reorganizing paragraphs, clarifying definitions, and refining figure captions). Substantive claims, citations, and reported results were authored and verified by the authors.

LLMs did not generate experimental results, design the core method, or serve as authors. The authors take full responsibility for all content, including any LLM-assisted text, and verified the accuracy of all citations and empirical results.

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

# A APPENDIX

## A.1 NOTATIONS

**Parameters**

| | |
|---|---|
| $\mathcal{K}$ | The size of the action space |
| $S$ | Side length of the cubic region of interest (ROI) |
| $L$ | Number of hops for ego-graph neighborhood |
| $T$ | Length of input sequence or total timesteps |
| $H_b$ | Number of sectors for spatial binning |

**Sets and Space**

| | |
|---|---|
| $\mathcal{G}^t$ | Embodied graph at time $t$ |
| $\mathcal{V}^t$ | Set of nodes at time $t$ |
| $\mathcal{E}^t$ | Set of edges at time $t$ |
| $\mathcal{I}^t$ | Set of interactions at time $t$ |
| $\mathcal{V}_a$ | Set of agent nodes |
| $\mathcal{V}_o$ | Set of object nodes |
| $\mathcal{E}_{oo}$ | Object-object edges |
| $\mathcal{E}_{oa}$ | Object-agent edges |
| $\mathcal{E}_{aa}$ | Agent-agent edges (multi-agent) |
| $\mathcal{I}_{oo}$ | Object-object interactions |
| $\mathcal{I}_{oa}$ | Object-agent interactions |
| $\mathcal{I}_{aa}$ | Agent-agent interactions |
| $\mathcal{X}_o, \mathcal{X}_a$ | Node features for objects and agents |
| $\mathcal{S}_{oo}, \mathcal{S}_{oa}, \mathcal{S}_{aa}$ | Edge features |
| $\mathcal{R}_{oo}, \mathcal{R}_{oa}, \mathcal{R}_{aa}$ | Interaction features |
| $\mathcal{N}_v^t$ | $L$-hop neighbors of node $v$ at time $t$ |
| $\mathcal{G}_v^t$ | Ego-embodied graph of node $v$ at time $t$ |
| $M$ | Grid map in ROI index space |

**Functions and Models**

| | |
|---|---|
| $f_\theta$ | Learnable dynamics network |
| $q_\phi$ | Encoder in world model |
| $p_\phi$ | Dynamics predictor |
| $\text{sg}(\cdot)$ | Stop-gradient operator |
| $r_{x,y,z}^{\text{walk}}$ | Walkable region condition |

**Losses and Objectives**

| | |
|---|---|
| $\mathcal{L}(\phi)$ | Total high-level world model loss |
| $\mathcal{L}_{\text{pred}}$ | Prediction loss |
| $\mathcal{L}_{\text{dyn}}$ | Dynamics loss |
| $\mathcal{L}_{\text{rep}}$ | Representation loss |
| $\mathcal{L}_{\text{graph}}$ | Graph reconstruction loss |
| $\mathcal{L}_{\text{rew}}$ | Reward prediction loss |
| $\mathcal{L}_{\text{con}}$ | Continuity loss |
| $\beta_{\text{pred}}, \beta_{\text{dyn}}, \beta_{\text{rep}}$ | Loss coefficients |

**Metrics**

| | |
|---|---|
| SR | Success Rate |
| SL | Success Length |
| SVR | Survival Rate |

**Indices and Variables**

| | |
|---|---|
| $t$ | Time step |
| $v, n$ | Node index |
| $x, y, z$ | Spatial coordinates |
| $c_n$ | Contact length of node $n$ |
| $V_n$ | Volume of node $n$ |
| $d_n$ | Distance from agent to node $n$ |
| $d_y^1$ | Vertical depth term for digging |
| $s_n$ | Availability score of node $n$ |
| $t_n$ | Type of node $n$ (frontier or digband) |
| $b^{\text{air}}, b^{\text{liq}}, b^{\text{diq}}, b^{\text{unk}}$ | Voxel type indicators |

**Miscellaneous**

| | |
|---|---|
| $A$ | Macro-action plan |
| $a$ | Primitive action |
| $h_t$ | Hidden state at time $t$ |
| $z_t$ | Latent state at time $t$ |

