# OpenReview forum: "Embodied Graph: A Relational World Model"
_ICLR.cc/2026/Conference — ICLR 2026 Conference Desk Rejected Submission_

### Official Review · Reviewer_JVD8 · 2025-10-21

**Soundness:** 3
**Presentation:** 2
**Contribution:** 2
**Rating:** 4
**Confidence:** 4

**Summary:**

This work presents a methodology to use embodied graphs to represent information about the environment and use it as an abstraction for world modeling. The idea is motivated by the fact that often pixel-based world modeling may contain a lot of irrelevant information, which can be unnecessary for task execution, and can hinder the agent’s abilities. By leveraging the embodied graph, this work proposes to use a relational world model, which has a higher-level embodied graph constructor that updates the graph for every time step based on agent observation, and a lower-level module that can execute actions directly in the environment.

**Strengths:**

- Graphs have often been brought up in the literature as useful signals to model environmental dynamics, given the structure and compactness of information they provide. This work makes another attempt at it and shows that, as compared to DreamerV3 (which does world modeling in pixel space).
- To enable world modeling across the embodied graph, the authors have proposed changes in the recurrent state space model used in DreamerV3, which can be valuable for future approaches interested in using graphs as a latent for world modeling.

**Weaknesses:**

- Although the paper presents an interesting idea and shows positive results on Minecraft and Atari, it presents limited analysis of their results. Given that these are tasks in simulation, they are good for a proof-of-concept, but by themselves, without any analysis, it is difficult to understand how various parts of the proposed methodology can benefit general decision-making literature.
- It would’ve been interesting to know the effect of various choices on how the node representation affects the performance, what kind of relationships had more impact on the performance (for instance, for agent-object, object-object relationships) which one matters more.
- Some more concrete experimentation details, in the main paper or appendix, would’ve been useful, for instance, it seems voxel observations are used for Minecraft, but it was unclear why exactly, as opposed to RGB views.

**Questions:**

- For training the world model, it seems like various metrics are used to approximate the graph, and those are reconstructed. What were these metrics? Is the transformation from the graph to these metrics learned, or approximated using heuristics?
- Also, for training data, was there a database of trajectories to first learn a world model, and then it was applied to a downstream task? If yes, was the dataset constructed using simulator metadata? Or was it trained along with the task itself?

- Minor:
  - Related work pointer: A related work, not in world modeling, but might be a good pointer in your related work. https://arxiv.org/pdf/2212.01186, they construct a scene graph as a supervisory signal for representations for object navigation.
  - Typos:
     - L177: section 3.1, it should be Fig 3, not Fig 3.1 (at least as per the figure)
     - L035: budge -> budget
     - L053: Fig. 2 -> Fig. 3

---

### Official Review · Reviewer_Z8DU · 2025-10-24

**Soundness:** 2
**Presentation:** 2
**Contribution:** 2
**Rating:** 2
**Confidence:** 4

**Summary:**

This paper introduces Embodied Graph, a relational world model that uses sparse, graph-based supervision to capture task-relevant dynamics for embodied agents. It addresses the problem of imagination drift caused by dense pixel-level supervision in traditional world models. The approach builds a dynamic graph of agent–object interactions and uses it to generate high-level macro-actions for exploration. Evaluated on Minecraft and Atari-100k, it outperforms strong baselines like DreamerV3, improving both survival rate and sample efficiency.

**Strengths:**

- The authors propose Embodied Graph, a novel world model architecture that mitigates error compounding through sparse, relational representations.
- RWM demonstrates strong sample efficiency and fast convergence on challenging tasks in Minecraft and Atari.

**Weaknesses:**

- The writing of this paper needs improvement. The terminology is inconsistent and occasionally confusing (e.g., “low-level RWM” vs. “low-level behavior model”).

- The approach relies heavily on handcrafted priors (e.g., voxel semantic masks, walkable region computation), which appear tailored to Minecraft and may not generalize to other domains or scale to multi-task settings.

- Macro-actions are defined as discrete interactions with graph nodes (e.g., dig, walk), raising concerns about applicability in continuous or fine-grained action spaces (e.g., precise robotic grasping).

- The paper omits any analysis of computational cost—dynamic graph construction, KNN edge generation, and macro-action scoring could be expensive in high-dimensional environments.

- Despite claiming to be an **embodied** world model, evaluation is limited to simulated games; experiments on physical robot benchmarks are missing.

- Key baselines are absent, notably [1], which also uses a hierarchical world model and has been evaluated on Minecraft.

[1] Open-World Reinforcement Learning over Long Short-Term Imagination. ICLR’25.

**Questions:**

- How can the reliance on Minecraft-specific priors be reduced to enable broader generalization?

- Can the RWM support multi-task training or larger-scale environments without redesigning the graph constructor?

- How would the macro-action design extend to continuous or high-DoF action spaces?

- What is the computational overhead of dynamic graph construction and planning?

- How does RWM perform on robotic benchmarks?

- How does RWM compare to recent hierarchical world models?

---

### Official Review · Reviewer_T4Mv · 2025-10-31

**Soundness:** 2
**Presentation:** 2
**Contribution:** 2
**Rating:** 2
**Confidence:** 3

**Summary:**

This paper proposes the Relational World Model (RWM), a novel graph-based learning paradigm for modeling environment dynamics in embodied exploration, overcoming the "imagination drift" of traditional world models. The RWM uses a hierarchical architecture, with a high-level model guided by a hand-crafted "Embodied Dynamic Graph Constructor" generating macro-actions, and a low-level plug-and-play policy model executing these actions. Experiments show the method significantly outperforms baselines, notably being the only one to collect diamonds in Minecraft.

**Strengths:**

1. The introduced paradigm of using graph in modeling abstracted environment dynamics is novel, and the proposed hierarchical architecture is reasonable.
2. The reported performance on the "collect diamond" task in Minecraft of the method is impressive.

**Weaknesses:**

1. The paper claims to solve the problem with a "sparse-supervision design". The actual method does the opposite: it uses a complex, hand-crafted parser to process dense voxel observations using a dense set of human-engineered rules (defining walkable region, frontiers, diggable regions, etc.). This is not "sparse supervision"; it is "dense, structured, human-prior supervision."
2. The graph constructor in Sec 3.1.1 is exclusively designed for 3D voxel data . The paper provides no explanation for how the "Embodied Graph" is constructed for the 2D pixel data in the Atari games.
3. The proposed method, RWM, uses the Dreamer V3 baseline as its low-level execution module. The comparison in Table 1 is therefore (RWM Graph Planner + Hand-Crafted Parser + Dreamer V3) vs. Dreamer V3. This is only a demonstration that adding a complex, bespoke planning module improves a base policy, which is an expected outcome.
4. The core of the method's success appears to be the "Dynamic Embodied Graph Constructor", which is a massive, non-trivial, non-general piece of domain-specific engineering for Minecraft. This is not a general learning paradigm.

**Questions:**

Could you please provide the methodology for how the Embodied Graph was constructed for the 26 2D pixel-based Atari games? This is entirely missing from the paper.

---

### Official Review · Reviewer_vSNA · 2025-11-03

**Soundness:** 3
**Presentation:** 1
**Contribution:** 2
**Rating:** 2
**Confidence:** 3

**Summary:**

This paper presents a hierarchical graph-based model for world modeling. The authors introduce an Entity-Interaction graph model that captures the high-level structure and dynamics of the world. In this way, they design a dynamics function that learns the evolution of their RWM. They model their learning problem based on DreamerV3-inspired losses. Finally, they evaluate RWM in Minecraft and Atari and achieve results that improve over DreamerV3 in very challenging tasks.

**Strengths:**

The paper introduces a new hierarchical graph-based for world modeling that proves empirically useful for very complex in the Minecraft (and Atari) and beats very powerful agents like DreamerV3 and they provide extensive evaluation in both the domains.

**Weaknesses:**

- Notation is hard to follow because of the lack of symbol and loss definitions.
- The paper needs some proof-reading. Apart from typos, there are sentences that seemed to have been left behind while editing. For instance, lines 314-315.
- Losses are not fully defined: what is the graph loss, for instance?
- They use Dreamer’s RSSM for the high-level model: what is embedding and how is the graph embedded?
- What is the high-level macro action? How is it embedded? Is the set of action fixed?
- It seems that RWM requires the definition of the entity graph with hand-designed feature extraction to be able to work with complex observations before the actual learning process. Moreover, it seems that the skills need to be provided too? It is unclear to me at this point how the macro-actions are designed/learned.
- Missing related works. There are relevant works in hierarchical graph-based decision-making that is missing and that touches in many ideas that seem relevant to this paper. Some of them are: (1) Bagaria et al. Intrinsically Motivated Discovery of Temporally Abstract Graph-based Models of the World, (2) Bagaria e at. Skill Discovery for Exploration and Planning using Deep Skill Graphs. I recommend the authors to further improve their literature review.
- The appendix doesn't seem to include any extra information about their model that would allow people to reproduce their work.

Overall, I believe the paper would benefit from extra polishing. I believe that their work shows some interesting ideas that need to be placed effectively within the literature and to improve its presentation. The results seem promising and, though it seems that it is very overfitted to the domains, I believe that I could be an effective model that is not fully generalizable yet but could lead to interesting future work.

**Questions:**

- What are the graph reconstruction metrics? They don’t seem to be defined anywhere.
- What are the low-level policies? Where do they come from?
- Is the number of nodes and type of nodes fixed beforehand?
- What is the high-level macro action? How is it embedded? How does it relate to the low-level actions/policies?
- I don’t fully understand what the macro-action generator actually does? Does it learn new macro-actions? Or it’s only the high-level decision making policy? Is this learned?

---

### Note · Program_Chairs · 2026-01-17
**Submission Desk Rejected by Program Chairs**

The following references in this submission do not refer to real documents and/or have major errors in bibliographic information:

 Hanxiao Li, Jiahui Xu, Xinlei Chen, Jianfeng Wang, Lingxi Xie, Chenyang Wu, and Yuke Zhu. Graphformer: Graph-based transformer architectures for embodied navigation. In Advances in Neural Information Processing Systems (NeurIPS), 2023.

Yuxiang Wang, Qifan Wu, Amanpreet Singh, Yonatan Bisk, and Ruslan Salakhutdinov. Adversarially learned knowledge graph for commonsense reasoning in embodied agents. In Advances in Neural Information Processing Systems (NeurIPS), 2021.

Siyuan Huang, Chenyang Wu, Huazhe Xu, and Yuke Zhu. A closer look at embodied navigation: Planning, navigation, and mapping with semantic graphs. In Conference on Robot Learning (CoRL), 2022.

Devendra Singh Chaplot, Dhiraj Gandhi, Abhinav Gupta, and Ruslan Salakhutdinov. Objectoriented slam using graph neural networks. In Advances in Neural Information Processing Systems (NeurIPS), 2020.